# The Impact of Head-Up Tilt Sleeping on Orthostatic Tolerance: A Scoping Review

**DOI:** 10.3390/biology12081108

**Published:** 2023-08-09

**Authors:** Amber H. van der Stam, Sharon Shmuely, Nienke M. de Vries, Bastiaan R. Bloem, Roland D. Thijs

**Affiliations:** 1Department of Neurology, Donders Institute for Brain Cognition and Behavior, Radboud University Medical Center, 6500 HB Nijmegen, The Netherlands; amber.vanderstam@radboudumc.nl (A.H.v.d.S.); sharon.shmuely@radboudumc.nl (S.S.); nienke.devries@radboudumc.nl (N.M.d.V.); bas.bloem@radboudumc.nl (B.R.B.); 2Department of Neurology, Leiden University Medical Centre, 2300 RC Leiden, The Netherlands; 3Stichting Epilepsie Instellingen Nederland, 2130 AM Hoofddorp, The Netherlands; 4UCL Queen Square Institute of Neurology, University College London, London WC1N 1PJ, UK

**Keywords:** autonomic failure, supine hypertension, orthostatic hypotension, nonpharmacological interventions, Parkinson’s disease, nocturia

## Abstract

**Simple Summary:**

Symptoms such as light-headedness and fainting upon standing can have a large negative impact on the quality of life, especially for people with orthostatic hypotension (blood pressure drop upon or during standing). One treatment option suggested in the clinic is head-up tilted sleeping (HUTS), where the full body is inclined. In this paper we reviewed the available evidence for the use of HUTS. We identified 10 studies focussing on HUTS as a treatment to improve orthostatic tolerance. Unfortunately, the overall evidence was weak, mainly because of the low number of included participants. We also noticed that the studied angles differed as well as the type of measurements to evaluate HUTS. Despite this, the anecdotal evidence suggested that HUTS therapy could slightly improve low standing blood pressure and its associated symptoms. The effects were more marked if higher angles were applied. These results provide some, although weak, evidence favouring HUTS, but the clinical relevance and the tolerability need to be studied further in larger-scale trials.

**Abstract:**

To systematically summarize the evidence of head-up tilt sleeping (HUTS) on orthostatic tolerance, we conducted a systematic, predefined search in PubMed, OVID Embase, Cochrane and Web of Science. We included studies assessing the effect of HUTS on orthostatic tolerance and other cardiovascular measures and rated the quality with the American Academy of Neurology risk of bias tool. We included 10 studies (*n* = 185) in four groups: orthostatic hypotension (OH; 6 studies, *n* = 103), vasovagal syncope (1 study, *n* = 12), nocturnal angina pectoris (1 study, *n* = 10) and healthy subjects (2 studies, *n* = 58). HUTS duration varied (1 day–4 months) with variable inclinations (5°–15°). In two of six OH studies, HUTS significantly improved standing systolic blood pressure. Orthostatic tolerance was consistently enhanced in OH studies with higher angles (≥12°), in 2 out of 3 with smaller angles (5°) but also in one studying horizontal sleeping. In vasovagal syncope, HUTS significantly augmented resilience to extreme orthostatic stress. One study was rated as a class II risk of bias, one of Class II/III and eight of Class IV. The evidence favouring HUTS to improve orthostatic tolerance is weak due to variable interventions, populations, small samples and a high risk of bias. Despite this, we found some physiological signs suggesting a beneficial effect.

## 1. Introduction

Orthostatic hypotension (OH) is an unusually large decrease in blood pressure (BP) upon standing and a very common physical sign, particularly among the elderly [1]. Causes can be neurogenic, e.g., synucleinopathies such as Parkinson’s disease, or non-neurogenic, e.g., drug-induced OH [2,3,4]. OH signifies the failure of compensational mechanisms (the fast baroreflex and the slower humoral activation) that are normally activated during sudden and prolonged orthostatic stress to maintain normotension against the effects of gravity while standing upright. OH has various clinical expressions, ranging from orthostatic intolerance (i.e., symptoms of presyncope while upright that are relieved when sitting or lying down) to unexplained falls and syncope [4,5]. As such, OH represents a significant clinical problem, as it is often associated with great disability and it may lead to debilitation and costly complications such as fall-related fractures or other injuries.

OH management primary consists of lifestyle advices such as standing with the legs crossed or increasing salt and water intake [4]. Pharmacological options are available for selected individuals, yet carry an important disadvantage as the BP increases, regardless of the body’s position. This is especially problematic in people with OH and an accompanying supine hypertension, which typically contributes to the long-term risk of adverse cardiovascular events in OH [6]. Sleeping in a head-up tilt position (HUTS) is a non-pharmacological intervention that not only alleviates symptomatic OH, but additionally does not worsen (and perhaps even improve) supine hypertension [7,8].

Although theoretically very attractive, the concept of HUTS is thus far merely based on several small-scale cohort studies and expert opinion [9]. Despite this lack of rigorous evidence, HUTS has been proposed as an effective and even first choice non-pharmacological treatment for OH for over three decades, for example, in international guidelines [10,11,12]. It is, however, often not recommended by clinicians in daily practice because of a lack of evidence on its effectiveness, the presumed poor tolerability by patients, and lack of concrete advice on how to implement this intervention.

With this scoping review, we aimed to systematically identify and summarize all relevant literature on the effect of HUTS on cardiovascular function, to improve our understanding of the mechanisms of action underlying HUTS, and to identify knowledge gaps that may guide future research.

## 2. Methods

### 2.1. Search Strategy and Selection Criteria

We used the scoping review method to identify and summarize all relevant literature [13,14]. We followed the 2018 preferred reporting items for systematic reviews and meta-analyses extension for scoping reviews while preparing the study protocol and study report [15]. We conducted a systematic search of PubMed, OVID Embase, Cochrane and Web of Science on 12 January 2023, using a combination of MeSH/EMTREE terms and key words (Appendix A).

We included (all criteria had to be met) the following:(1)Studies of people with or without autonomic dysfunction;(2)Studies of people aged ≥6 years;(3)Articles assessing the effect of full-body head-up tilt sleeping of any angle;(4)Articles with outcome measures related to cardiovascular control (e.g., orthostatic tolerance, BP, weight, oedema and nycturia).

We excluded the following:(1)Studies simultaneously evaluating HUTS with other pharmacological treatments for OH, including salt loading;(2)The following article types: case reports, narrative reviews, expert opinions, editorials, design studies and systematic reviews.

We did not exclude studies based on publication language, but arranged for translation. If multiple articles were based on the same study data, we included the most complete report not to overrepresent the data. We included articles with any number of participants and of any quality or study design. We used Rayyan to screen the records (rayyan.ai/). We manually searched the bibliographies of all included studies for potentially relevant studies. We also checked the bibliography of all excluded systematic reviews.

### 2.2. Study Selection on Data Extraction

Two reviewers (S.S. and A.S.) independently screened all titles and abstracts identified by the initial search. Next, we obtained the full texts of any article deemed possibly relevant by either reviewer These full texts were then independently evaluated by two reviewers (R.D.T. with S.S. or A.S.) to decide whether the study was to be included. Disagreements were settled by consensus.

One reviewer (S.S.) extracted the data from each study using a form specifically designed for this review, including author(s), year of publication, study type, source population, sample characteristics (i.e., age, sex and cardiovascular medication), HUTS characteristics (e.g., angle(s), duration), OH definition, details of OH assessment (e.g., time of day, salt and fluid intake), and all cardiovascular outcome measures.

The relevant outcome measures to evaluate the impact of HUTS depend on the studied population. In people with OH, a beneficial effect of HUTS would translate to an amelioration of orthostatic tolerance, a higher standing BP and lower orthostatic BP drop. In those with OH combined with supine hypertension, we would also expect a lower supine BP. The aetiology of OH may also be relevant when evaluating HUTS as the mechanisms differ and disease courses may vary. Healthy people or cases with vasovagal syncope (i.e., a form of reflex syncope due to a specific set of emotional or orthostatic triggers) [5] have well-functioning compensatory mechanisms to maintain normotension in normal conditions. Therefore, little to no change in BP due to HUTS is expected. These subjects may, however, experience improved orthostatic tolerance for extreme orthostatic stress (i.e., longer time to syncope) or a reduction of the physiological BP perturbations in the first 30 s of active standing [4]. We therefore evaluated various BP parameters and selected the relevant ones depending on the study population.

### 2.3. Applied Methods

We selected a total of 16 study parameters for assessing the methodological quality of HUTS studies. Eight of these items were applicable to all studies, i.e., reporting of duration, angle, tolerance and compliance of HUTS, quantitative evaluation of orthostatic symptoms, nocturia volume and overnight body weight change. Six parameters related to the circumstances of orthostatic BP measurements, i.e., sufficient duration of supine rest ≥5 min and standing time ≥3 min, report of similar time of day of measurements, hydration and fasting state, and before or after drug administration. Only two of these were applicable to OH populations, namely aetiology (neurogenic vs. non-neurogenic OH) and the presence of supine hypertension (defined as systolic BP ≥ 140 mmHg and/or diastolic BP ≥ 90 mmHg after ≥5 min of supine rest) [16]. We counted the proportion of reported applicable parameters for each study.

### 2.4. Risk of Bias

We rated the risk of bias of each included article using the American Academy of Neurology (AAN) risk of bias class of evidence scheme for therapeutic studies, also known as the level of evidence [17]. In this scheme, studies rated as Class I are judged to have a low risk of bias; Class II, a moderate risk of bias; Class III, a moderately high risk of bias; and Class IV, a very high risk of bias. Two reviewers (S.S. and A.S.) independently assessed the risk of bias of each study. Disagreements were settled by consensus.

### 2.5. Data Analysis

Descriptive statistics were used to present the results. To illustrate the effect size of HUTS on the orthostatic systolic BP values (supine, standing and BP change upon standing) in patients with OH, we calculated the mean, SE, and 95% confidence intervals of the difference between the post- vs. pre-HUTS values and created a forest plot. We were unable to perform a formal meta-analysis due to the heterogeneous interventions (e.g., HUTS angle or duration), populations and outcomes.

## 3. Results

### 3.1. Selection of Sources

We identified 773 studies with our initial search (Figure 1). We excluded 739 studies after screening the titles and abstracts and assessed 29 reports for eligibility. Of these, we included six articles [18,19,20,21,22,23] and two meeting abstracts [24,25]. After reviewing the references of the included studies, we included two additional articles [26,27].

### 3.2. Study Protocols and Populations

Characteristics of the 10 included articles assessing the effect of HUTS on cardiovascular control are shown in Table 1. A total of 185 people underwent HUTS at different angles and with different durations. Study types were prospective cohort studies (*n* = 6), case series (*n* = 2), a cross-over trial (*n* = 1) and a randomised controlled trial (*n* = 1). Studied populations included OH (*n* = 6; a total of 103 cases undergoing HUTS and 34 OH cases in a placebo group), vasovagal syncope (*n* = 1; 12 cases), healthy people (*n* = 2; 58 cases) and people with angina pectoris (*n* = 1; 10 cases). Five out of six OH studies provided some clinical details to at least partially differentiate between neurogenic and non-neurogenic OH. The authors of [24] specifically targeted a population with Parkinson’s disease and OH. The one RCT did not provide information on the aetiology [23].

### 3.3. Methodological Quality

Table 2 shows the score of study parameters for assessing the methodological quality of HUTS studies for each of the included studies. Six OH studies could score a maximum of 16 points, two non-OH studies measuring orthostatic BP could score a maximum of 14 points and two non-OH studies that did not perform orthostatic BP measurements could score a maximum of eight points. The median score of the 10 included studies is 37%, ranging from 25% to a maximum of 68%.

### 3.4. HUTS Implementation

Five of 10 studies applied HUTS at home [20,21,23,24,25], two in the hospital [18,22] and three started in the hospital and had a follow-up at home [19,26,27]. HUTS implementation varied among studies with variable tilting angles (median = 6° (5° to 15°)) as well as various durations (median = 7 days (1 day to 6 months)) (Figure 2). There was one randomized controlled trial, which compared 5° HUTS (*n* = 66) versus no HUTS (*n* = 34) in a total of 100 people with symptomatic OH [23].

Several different HUTS application methods were used. Some used blocks or chairs underneath the head of the bed (*n* = 3) [21,23,27], some used wedge mattresses (*n* = 1) [25], or an adjustable hospital bed (*n* = 1) [18], one study had HUTS implemented at home by an engineer (*n* = 1) [20], and one used home-built tools (*n* = 1) [19]. Three studies did not specify the method [22,24,26].

A pillow underneath the mattress at the height of the thighs is the most commonly deployed preventative method to keep patients from sliding down (*n* = 3) [19,22,27]; two studies reported the use of a footboard with optional pillows to prevent foot pain (*n* = 2) [19,22] and one study mentioned the use of a sleeping bag attached to the headboard of the bed (*n* = 1) [19]. The remaining seven studies did not mention the use of any precautions.

### 3.5. Orthostatic Hypotension Definition

Only two of the six studies of OH populations specified the definition of OH. Fan and colleagues (2009 and 2011) utilized the 1996 consensus statement of the American Autonomic Society and the American Academy of Neurology (i.e., systolic BP decrease of ≥20 mmHg, or a diastolic BP decrease of ≥10 mmHg, within 3 min after changing from a supine to standing position) [29]. This definition matches the 2011 consensus statement, which adds that supine rest before head-up tilt or standing up should be last at least 5 min and that in patients with supine hypertension, a decrease in systolic BP of ≥30 mmHg is required [30]. The two studies of Fan and colleagues did not report baseline supine BP values and therefore it is unknown whether any of the cases had supine hypertension. The other four studies did not define OH [19,24,26,27]. When studying the data of these four studies, however, it seems that three cases do comply with the abovementioned 1996 consensus statement. Only for one study was this not completely certain as only the mean values are provided for supine BP as well as for the BP drop at baseline (orthostatic drop systolic BP 27 ± 20 mmHg; diastolic BP 16 ± 15 mmHg; mean morning systolic BP 101 ± 25 mmHg; diastolic BP 67 ± 17 mmHg) [24].

### 3.6. Tolerance

Tolerance was reported in 6 of 10 studies. HUTS was tolerated well by all nine patients in one of the low-angle studies (5°) [22]. The other five studies reporting on tolerance did not quantify this parameter. During HUTS of 12–13°, problems with tolerance were noted, with the most common complaints being sliding down [19,21] or stiff legs from leg oedema [21,26,27]. The study that used the steeper angle of 15° for one night noted that it was well tolerated in this healthy population, which was supported by an unchanged sleep time pre- vs. post-HUTS (380 ± 14 min vs. 375 ± 15 min), as scored automatically by a clinically validated home-sleep test [25].

### 3.7. Compliance

Only one of five completely home-based studies evaluated compliance, reporting a self-reported compliance of 77% (HUTS 5° for six weeks) [23]. Three studies did not investigate compliance yet reported a long-term home-based follow-up of HUTS (*n* = 9, 2–70 months) which may serve here as an indirect marker [19,26,27].

### 3.8. Main Findings

#### 3.8.1. Orthostatic Blood Pressure

Eight studies conducted orthostatic BP measurements of OH (*n* = 6), vasovagal syncope (*n* = 1) and healthy populations (*n* = 1). The methods used and details of the assessments are given in Table 1. We summarised the effect of HUTS on orthostatic systolic BP in the six OH studies (Figure 3). We could calculate mean difference pre- vs. post-HUTS and confidence intervals of standing systolic BP in five studies and systolic BP difference upon standing in four studies. Only a few studies reported a significant difference following HUTS. Although all mean effect sizes were favouring of HUTS, in the RCT, the mean increase in standing systolic BP following HUTS with a low HUTS angle (5°) did not significantly differ from horizontal sleeping [23].

In the vasovagal syncope population, the resilience to prolonged tilting with additional graded lower body negative pressure improved after three to four months of HUTS at an angle of 10°. In 11/12 cases (92%), time to pre-syncope improved (mean increase of 7.8 ± 1.6 min; after) [20]. In the healthy population, the mean Δ systolic BP drop after 10 s of active standing reduced following HUTS without impacting the nadir systolic BP at two minutes [21].

#### 3.8.2. Orthostatic Symptoms and Syncope

Orthostatic symptoms were reported in five of the OH studies. In three OH studies, all cases (total *n* = 10) reported an amelioration of orthostatic symptoms (HUTS angles 12–13°) [19,26,27], one study reported improved symptoms in six of nine individuals (HUTS angle 5°) [22], and the last study (RCT) reported a significant improvement of symptoms of dizziness per week in the HUTS (5°) (*n* = 66, *p* = 0.0039) but this was also significant in the non-HUTS group (*n* = 34, *p* = 0.0013), and there was no difference between the groups (*p* = 0.27) [23]. During long-term follow-up of 2–4 months, three out of four cases reported that no more syncope had occurred [26]. Two of these cases discontinued HUTS for a short period to investigate whether HUTS truly reduced the symptoms. In both cases, orthostatic intolerance returned supported by worsening of the orthostatic blood pressure and the return of symptoms within two days [26,27].

Among the 12 subjects with vasovagal syncope, 11 cases (92%) reported a reduction in presyncope following HUTS [20]. In the 29 healthy subjects, HUTS significantly lowered the incidence of light-headedness during an active standing test (from 93.1% to 41.4%) [21].

#### 3.8.3. Other Blood Pressure Data

Three OH studies [22,23,24] and one study on healthy subjects [21] conducted 24h ABPM and found no significant change in mean overall, day- or night-time BP before and during HUTS (5–6°). None of these studies reported the presence of supine hypertension.

The study on nocturnal angina reported a significant decrease in central venous pressure and diastolic pulmonary artery pressure during whole body HUTS (10°) compared to the control night (with only the head up) [18].

#### 3.8.4. Other Variables

One of the mechanisms through which HUTS may ameliorate BP control is the increase in volume and a redistribution of body fluids. Three studies monitored plasma volume, all reporting an increase after HUTS (Table 3). One study showed that the blood volume increased after 3 to 4 months of HUTS, in six of eight cases with vasovagal syncope (average 3.18 L/kg to 3.40 L/kg). This increase correlated with the prolonged time until syncope after tilt and lower-body negative-pressure application. The two cases with vasovagal syncope without increased plasma volume showed no or only a very limited increase in orthostatic tolerance [20]. Two OH studies measured the blood volume measured in two cases: both had a higher blood volume following HUTS (increase of 0.6 litres in one [27]; 6 cc/kg in another [26]).

Five out of the ten included studies monitored changes in body weight following HUTS (Table 3), one overnight weight loss and three urinary output. A total weight gain could indicate better fluid retention but could also be explained by many other factors. The overnight weight is a more specific marker reflecting the amount of fluid lost over-night, with larger fluid depletion thought to increase the severity of OH in the morning. Overall, within the OH and healthy population, HUTS resulted in either an increase in weight [21,27] or did not influence weight [22,23]. One OH study using 12° HUTS showed that the average weight lost during the night did not change, even though total weight did increase [19]. In three studies, the urine output (volume status and concentration) (Table 3) was evaluated; in all studies, participants were required to have an intake of at least 2 litres of fluid during the day. Two of the studies focussing on an OH patient group split the urine collection into a day and night sample. One study found a non-significant increase in the day/night ratio of sodium excretion, reflecting a lower excretion at night [19]. Urinary volume was only discussed in one other OH population where the night-time volume was significantly reduced by 145 mL after 6 days of HUTS [21]. The daytime volume did not change, and neither did night- nor day-time sodium excretion [21]. None of the studies had nocturia as an outcome measure.

Water retention and a more upright position may lead to ankle oedema, and this was measured in four studies: three with OH and one with a healthy population (Table 3). One study measured ankle circumference both before and after HUTS in a healthy population and reported an increase in ankle circumference of 8 mm following 6 days of 12° HUTS [21]. The other studies encompassed two case studies where the individuals had slight pitting oedema after 3 and 4 days of HUTS [26,27]. A study in the OH population reported an increase in oedema to 41% in the HUTS group, compared to 19% in the non-HUTS group but did not specify the applied method [23]. Additionally, blood laboratory analyses were performed in four studies with varying outcome measures (Table 3).

## 4. Discussion

This systematic scoping review of the impact of HUTS on orthostatic tolerance identified a small number of studies, collectively showing weak but consistent evidence of a potential positive effect of this non-pharmacological intervention. The 10 included studies were mostly cohort studies with small sample sizes, with a high risk of bias that included heterogenous study populations, a variable HUTS implementation (i.e., angles and duration) and a range of OH assessment methods. The overall methodological quality score, based on a total of 16 parameters including compliance and tolerance of HUTS, was very low.

### 4.1. Summary of Evidence

Our primary interest was the effect of HUTS on orthostatic blood pressure in populations with OH. Most studies failed to categorise the OH type. It is likely, however, that those with neurogenic OH will profit most from HUTS as OH in this population is severe and mostly coincides with supine hypertension. Although there appeared to be a fairly consistent trend towards BP effects favouring HUTS in the diverse OH populations, most results did not reach significance, possibly due to the small sample sizes. The impact of HUTS on OH was more pronounced for those OH cases subjected to higher vs. lower HUTS angles, but the number of studied cases with high HUTS angles was lower, thus causing wider confidence intervals. We observed that the protocol for measuring OH varied greatly among the studies, which may have also impacted the analysis of the efficacy of HUTS. Often, only rather short periods of standing (<2 min) were applied to evaluate immediate OH, which may have hampered the identification of more long-term BP changes that are equally relevant in daily life. The circumstances of most OH measurements were not ideal as well. Most studies did not specify the time of day the orthostatic BP measurements were performed and whether the time was kept constant in both pre- and post-HUTS evaluation.

All persons with OH that were treated with high angles of HUTS and most persons treated with smaller HUTS angles reported less orthostatic symptoms. The placebo group of the one RCT, however, also reported significantly improved orthostatic symptoms, and this improvement did not differ from the HUTS group [23]. We speculate that, apart from the expectation effect, the natural course (over the 6-week treatment interval) may have explained the improvement as some forms of OH (particularly non-neurogenic OH) may be self-limiting. Another possible explanation for the improvement in the control group is the medical intervention itself: all people received information about the diagnosis and may have applied additional lifestyle measures. One practical recommendation for future studies is to predominantly include persons with longstanding OH that would therefore be unlikely to resolve spontaneously and the improvement that can be achieved here is the largest. At this moment, there is only one RCT available, and due to the nature of the intervention, control groups of a good quality will be difficult to create. A fully blinded control group is not achievable since, unlike in pharmacological interventions, a placebo cannot be given. Careful consideration must therefore be made on the precise composition of the control groups. Obviously, we must also consider the possibility that HUTS is not an effective treatment (and we are open to that option), but there are several arguments that would appear to argue against this.

Specifically, we found some physiological indications of a beneficial effect of HUTS. The improvement in orthostatic BP control among people with OH was more marked when comparing higher vs. lower HUTS angles, suggesting a dose–response effect that would be compatible with a genuine treatment effect (although the angle could understandably not be blinded). Also, although the angle studied was small, the HUTS group had more ankle oedema compared to the placebo group [23]. Ankle oedema indicates a redistribution of body fluids. It acts as a water jacket and was found to correlate with better orthostatic tolerance [31]. The incidence of oedema following HUTS is thus a physiological sign that may contribute to improved BP control in OH, although the inclination may have been too small to demonstrate efficacy [23]. Interestingly, we found some evidence that HUTS improves BP homeostasis in people with vasovagal syncope and healthy controls by increasing their resilience to extreme orthostatic stress [20,21]. Other physiological signs suggesting a beneficial effect of HUTS include the consistent trend towards increased volume and lower night-time urine [20,21,26,27]. HUTS has the unique potential to lower supine BP in people with neurogenic OH and coexisting supine hypertension. From a physiological perspective, one would even expect a more marked effect on supine hypertension rather than on orthostatic hypotension. We could, however, not evaluate this effect here as none of the studies reported the presence of supine hypertension. We recommend that assessment of supine hypertension should be routinely included in future evaluations of HUTS.

### 4.2. Strengths and Weaknesses of the Review

This is the first review to systematically synthesise the evidence for the treatment of orthostatic intolerance with HUTS. Our review included all population types of all ages and a broad range of outcome measures that relate to cardiovascular function. A limitation of the review is that we could not pool the findings as the interventions (angle and duration) varied extremely across the studies. We did not include studies that simultaneously studied the effect of HUTS with another non-pharmacological or pharmacological intervention and, therefore, we had to exclude potentially relevant studies analysing the HUTS intervention.

### 4.3. Future Directions

Although HUTS is an attractive and simple intervention, with the unique ability to positively impact both orthostatic hypotension and supine hypertension, it has not been widely adopted in daily clinical practice because of the lack of well-controlled studies that could guide such a clinical implementation. Future research should provide robust data on the clinical efficacy of HUTS, particularly in those with longstanding neurogenic OH and co-existing supine hypertension. The optimal tilt angle should be determined by studying the trade-off between tolerability and efficacy, which may vary among individuals. The minimal treatment duration that is needed to achieve a tangible clinical improvement also remains to be determined. Such future studies should be conducted in a home environment with BP evaluations, ideally complemented with standardised clinical evaluations of postural BP control. Outcomes should obviously be addressed towards blood-pressure control (OH and supine hypertension), but should also focus on other more long-term consequences, such as falls and fall-related injuries, or secondary vascular damage in the brain or elsewhere [4,32]. Long-term compliance also remains to be studied. Future studies are also needed to identify easy-to-access markers to predict a good clinical response and help optimize clinical implementation.

## 5. Conclusions

The evidence of the impact of HUTS on orthostatic tolerance is weak due to heterogeneous populations, variable HUTS angles, variable cardiovascular and other outcome measures, small sample sizes and therefore high risks of bias. Despite these limitations, we found some physiological signs suggesting a beneficial effect HUTS with more marked changes at higher angles. Yet the trade-off between HUTS efficacy and tolerability is the major unknown. Future well-controlled studies are needed to provide robust data of the clinical efficacy, optimal tilt-angles and tolerability.

## Figures and Tables

**Figure 1 biology-12-01108-f001:**
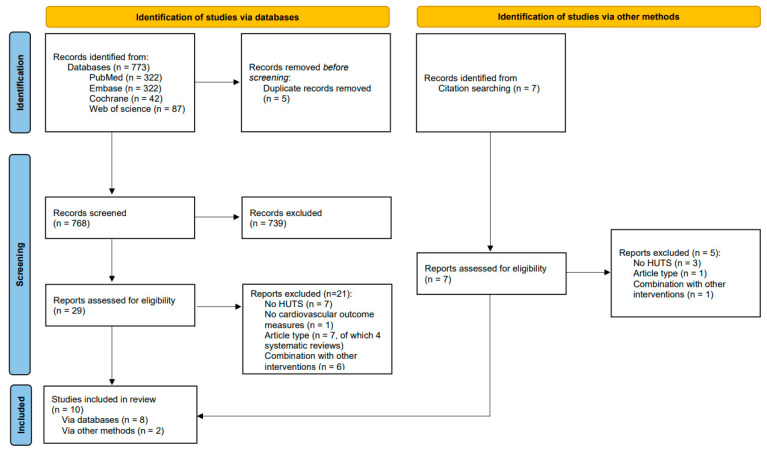
PRISMA flow diagram. HUTS = head-up tilt sleeping; PRISMA = preferred reporting items for systematic reviews and meta-analyses [28].

**Figure 2 biology-12-01108-f002:**
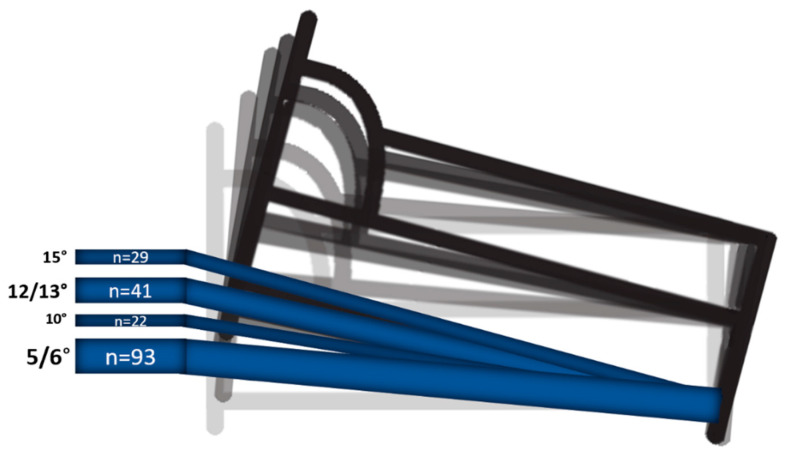
Illustration of the different angles of HUTS applied in the included studies. The number of cases subjected to the specific angles are indicated.

**Figure 3 biology-12-01108-f003:**
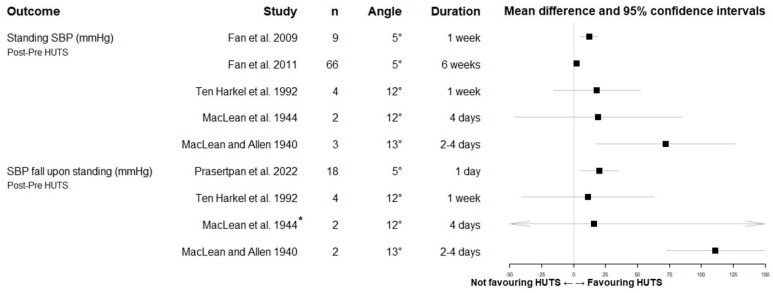
Forest plot showing mean differences and 95% confidence intervals of orthostatic systolic blood pressure (SBP) values (i.e., standing and change upon standing) after and before head-up tilt sleeping intervention (HUTS) in studies with orthostatic hypotension (*n* = 6; 102 cases). Favouring HUTS (towards the right) is a higher standing BP and smaller drop post HUTS, shown as the increase in the SBP change. Many of the included studies had a very limited sample size, resulting in unreliable estimations of the mean and confidence interval and a high likelihood of type II errors. In the case series, we only calculated the mean difference if we had access to the data of at least two cases [26,27]. * Values corresponding to this study: 95% CI—130 to 162 mmHg. SBP = systolic blood pressure; HUTS = head-up tilted sleeping.

**Table 1 biology-12-01108-t001:** Characteristics and results of all included studies (*n* = 10) studying the impact of head-up tilt sleeping (HUTS) on cardiovascular control.

First Author and Year	Study Type	Population	Cases *n*	Age (y; Mean (SD))	Female *n* (%)	HUTS Angle (°)	HUTS Duration	Collected Data	Method Orthostatic BP Measurement	Details of OH Assessment ^c^	Risk of Bias Class ^d^
Fan et al., 2009 [22]	Prospective cohort	Elderly with symptomatic OH of all causes	9	76 (5)	5 (55)	5	1 w	Orth. symptoms, orth. BP, ABPM, weight, lab	Active standing, beat-to-beat BP (Finapres). Supine 5 m; stand 120 s.	NR	IV
Fan et al., 2011 [23]	Randomised controlled trial	Elderly with symptomatic OH of all causes	100 - HUTS 66 - contr. 34	(Median, IQR) 76 (71, 80) 76 (72, 83)	37 (56) 19 (56)	5	6 w	Orth. symptoms, orth. BP, ABPM, weight, urine volume and Na, oedema	Active standing, beat-to-beat BP (Finapres). Supine 5 m; stand 120 s.	Both HUTS and non-HUTS group increased water intake to 2 L a day.	II
Prasertpan et al., 2022 ^a^ [24]	Prospective cohort	nOH in PD	18	69 (5.6)	11 (61)	6	1 d	Orth. BP, ABPM	NR	Morning immediately after awaking.	IV
Ten Harkel et al., 1992 [19]	Prospective cohort	nOH	4 ^b^	23; 44; 59; 65	3 (50)	12	1 w FU 8–70 m	Orth. symptoms, orth. BP, weight, urine K/Na/Creatinine	Active standing, beat-to-beat BP (Finapres). Supine 20 m; stand max 10 m or until symptoms.	At 08.00 h after an overnight fast. High salt intake of 150–200 mmol Na+/d and water intake of ≥2 L started 1w before HUTS.	IV
MacLean et al., 1944 [26]	Case series	Non-nOH	2	35; 57	0 (0)	12	4 d FU 3–6 m	Orth. symptoms, orth. BP, oedema, plasma volume, lab	Active standing. Supine before arising; stand various 1–25 m.	Before arising in the morning after overnight fast. Intake of water was controlled (not specified).	IV
MacLean and Allen 1940 [27]	Case series	nOH and non-nOH	4	59; 30; 34; 47	2 (50)	13	2–4 d FU (*n* = 3) 2–6 m	Orth. symptoms, syncope, orth. BP, oedema, plasma volume, lab, sweating	Active standing. Supine duration NR; stand 1–60 m or duration NR.	NR	IV
Cooper and Hainsworth 2008 [20]	Prospective cohort	VVS and poor orthostatic tolerance	12	42 (5)	6 (50)	10	3–4 m	Orth. symptoms, syncope, orth. BP, plasma volume	Orthostatic stress test: supine 20 m; tilt 60° for 20 m; lower body negative pressure until pre-syncope.	NR	IV
Fan et al., 2008 [21]	Prospective cohort	Healthy college students	29	22 (1.9)	16 (55)	13	1 w	Orth. symptoms, orth. BP, ABPM, oedema, weight, urine volume and Na, lab	Active standing, beat-to-beat BP (Finapres). Supine 5–10 m; stand 2 m.	Morning 9:00–11:00. Water intake of ≥2 L started 1 w before HUTS.	IV
Pham et al., 2019 ^a^ [25]	Cross-over	Healthy Peruvian highlanders	29	62.3 (8.9)	11 (38)	15	1 d	Sleep, respiratory variables, heart rate	NA	NA	II or III ^e^
Mohr et al., 1982 [18]	Prospective cohort	Refractory nocturnal angina	10	56.4 (4,8)	2 (20)	10	2 d	Aortic pressure, central venous pressure, pulmonary artery pressure	NA	NA	IV

^a^ Meeting abstract. ^b^ Six cases were studied; yet, in only four cases was HUTS the sole intervention; in the other two, HUTS was combined with fludrocortisone. ^c^ We searched the articles for further details of OH assessment including time of day, prior to assuming sitting position, fasting state, salt intake, before/after drug administration, hydration state and exercise. ^d^ As calculated using the American Academy of Neurology risk of bias tool [17]. ^e^ Meeting abstract contains insufficient information to classify. ABPM = ambulatory blood pressure measurement; BP = blood pressure; FU = follow-up; HUTS = head-up tilt sleeping; NA = not applicable; NR = not reported; (n)OH = (neurogenic) orthostatic hypotension; PD = Parkinson’s disease.

**Table 2 biology-12-01108-t002:** Score of study parameters for assessing the methodological quality of HUTS studies for each of the included studies. NA = not applicable; (n) OH = (neurogenic) orthostatic hypotension; SH = supine hypertension. Red (●) indicates the parameter was absent, green (●) indicates the described parameter was available in the study.

	Fan et al., 2009 [22]	Fan et al., 2011 [23]	Prasertpan et al., 2022 [24]	Ten Harkel et al., 1992 [19]	McLean et al., 1944 [26]	McLean and Allen 1940 [27]	Cooper and Hainsworth 2008 [20]	Fan et al., 2008 [21]	Pham et al., 2019 [25]	Mohr et al., 1982 [18]	Total Score (n)	Total Score (%)
**OH populations**												
Report of OH aetiology	●	●	●	●	●	●	NA	NA	NA	NA	5	83
Presence of SH mentioned	●	●	●	●	●	●	NA	NA	NA	NA	1	17
**Orthostatic BP protocol**												
Supine rest ≥ 5 m	●	●	●	●	●	●	●	●	NA	NA	6	75
Standing ≥ 3 m	●	●	●	●	●	●	●	●	NA	NA	2	25
Constant time of day	●	●	●	●	●	●	●	●	NA	NA	4	50
Accounting for hydration state	●	●	●	●	●	●	●	●	NA	NA	4	50
Accounting for fasting state	●	●	●	●	●	●	●	●	NA	NA	2	25
Before/after drug administration	●	●	●	●	●	●	●	●	NA	NA	0	0
**HUTS reporting**												
HUTS duration	●	●	●	●	●	●	●	●	●	●	10	100
HUTS angle	●	●	●	●	●	●	●	●	●	●	10	100
HUTS tolerance	●	●	●	●	●	●	●	●	●	●	6	60
HUTS compliance	●	●	●	●	●	●	●	●	●	●	1	10
Quantitative symptom evaluation	●	●	●	●	●	●	●	●	●	●	2	20
Nocturia: urine volume	●	●	●	●	●	●	●	●	●	●	2	20
Overnight ∆ body weight	●	●	●	●	●	●	●	●	●	●	1	10
Sleep quality	●	●	●	●	●	●	●	●	●	●	1	10
**Total score (n)**	5	7	4	11	9	4	4	7	4	2		
**Total score (%)**	31	43	25	68	56	25	28	50	50	25		

**Table 3 biology-12-01108-t003:** Other variables noted in the publications (*n* = 8). * Indicates significant change.

Variable	First Author and Year	Population (*n*)	Method	Outcome
Plasma volume, pre and post HUTS	Cooper and Hainsworth 2008 [20]	VVS (8)	Evans blue dye dilution method, 8 out of 12 cases	3.18 to 3.40 L/kg *
	MacLean et al., 1944 [27]	OH (1)	Unknown method, in 1 case	38.6 to 43.0 cc/kg
	MacLean and Allen 1940 [26]	OH (1)	Congo red method, in 1 case	45 to 51 cc/kg
Body weight, pre and post HUTS	Ten Harkel et al., 1992 [19] Fan et al., 2009 [22] Fan et al., 2011 [23] MacLean et al., 1944 [27] Fan et al., 2008 [21]	OH (4) OH (9) OH (100) OH (1) Healthy (29)	Measured post-voiding at 22:00 and 8:00 Unknown method Unknown method, controls compared to HUTS group Day before and after 3 days of HUTS, in 1 case Measured post-voiding at 8:00	Morning weight: 0.5 kg increase * Evening-morning difference: no change 70.0 to 70.7 kg No change 86.2 to 87.1 kg 66.1 to 66.5 kg *
Urine, Pre and post HUTS	Fan et al., 2008 [21] Fan et al., 2011 [23] Ten Harkel et al., 1992 [19]	Healthy (29) OH (100) OH (4)	Volume and sodium excretion 24 h volume and sodium excretion Creatinine, sodium, and potassium as day/night ratio	Night-time volume: 622 to 477 mL * Day-time volume: 1510 to 1562 mL Sodium excretion: 373 to 382 mmol Volume and sodium excretion: No change Creatinine and Potassium, no change. Sodium: 0.63 to 0.81
Oedema	Fan et al., 2008 [21] Fan et al., 2011 [23] MacLean et al., 1944 [27] MacLean and Allen 1940 [26]	Healthy (29) OH (100) OH (1) OH (1)	Measured calf and ankle circumference pre- and post-HUTS Unknown method Observation, 1 case Observation, 1 case	Ankle: 255 to 263 mm * Calf: 371 to 373 mm HUTS: 41%, controls: 19% * “slight pitting oedema” “slight oedema of the lower extremities”
Laboratory blood values Pre and Post HUTS	Fan et al., 2009 [22] Fan et al., 2008 [21] MacLean et al., 1944 MacLean and Allen 1940 [27]	OH (9) Healthy (29) OH (1) OH (4)	Haematocrit, plasma renin, electrolyte, aldosterone, creatinine Supine haematocrit, plasma renin, electrolytes, aldosterone, pro-ANP Haematocrit, chloride, protein Haematocrit, haemoglobin, and erythrocyte count	Creatinine: 101 to 95.6 mmol/L * All others: no change Haemoglobin 13.6 to 13.3 g/dL * All others: no change Haematocrit: 35.5% to 34.8% Chloride: 99.3 to 103.8 mEq/L Protein: 6.40 to 6.45 Gm/cL Haematocrit: 36% to 34% Haemoglobin: 8.5 to 8.1 Gm/cL Erythrocytes: 3.3 to 4.1 × 10^6^ per mL
Respiratory	Pham et al., 2019 [25]	Healthy (11)	Hypoxia burden during HUTS compared to flat sleeping	SpO_2_: 83.6% to 85.5% * RDI: 21.5 to 17.8/h *
Sleep	Pham et al., 2019 [25]	Healthy (11)	Total monitored sleep time, during HUTS compared to flat sleeping	Sleep time: 380 to 375 min

OH = orthostatic hypotension; HUTS = head-up tilted sleeping; SpO_2_ = nocturnal oxyhaemoglobin saturation; RDI = respiratory disturbance index.

## Data Availability

No new data were created or analysed in this study. Data sharing is not applicable to this article.

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
