# Peer review of "The Impact of Head-Up Tilt Sleeping on Orthostatic Tolerance: A Scoping Review"

_biology, 2023, doi:10.3390/biology12081108_

Round 1

Reviewer 1 Report

The present paper aimed to systematically identify and summarize all relevant literature on the effect of HUTS on cardiovascular function, to reveal the mechanisms of action underlying HUTS, and to identify knowledge gaps that may guide future research. 

A few changes are needed, as follows:

Introduction: Please define “orthostatic tolerance” , “orthostatic stress” and “vasovagal syncope”.

There are several other causes of orthostatic hypotension, such as prolonged immobility, venous pooling, physical exhaustion and endocrine disorders (Mozos I. Pathophysiology. Lecture Notes for Dental Medicine. LAP Lambert Academic Publishing. 2015). Please mention them, as well.

Reviewer 2 Report

This scoping review aimed to summarize the evidence behind head-up tilt sleeping (HUTS) on orthostatic tolerance. Accordingly, the authors performed a systematic search of research databases and ultimately included studies assessing the effect of HUTS on orthostatic tolerance and selected cardiovascular measure. In addition, they also rated the quality with the American Academy of Neurology risk of bias tool. After filtering out some studies by a predefined criteria, the review involved a total of 10 studies and 185 total subjects. These studies were divided into four groups: orthostatic hypotension (OH) 6, vasovagal syncope 1, nocturnal angina pectoris 1, and healthy subjects 2. The main finding was that the evidence for HUTS ability to improve orthostatic tolerance was weak as the current literature has had limitations such as different interventions studied, different populations, small sample sizes, and even a high risk of bias. Thus, overall the authors concluded that HUTS has beneficial effects, but they appear to be rather small, but this could be due to the limitations of current studies.

Overall, I think this review was well-done, on a very important topic, and provided a great deal of interesting information. This topic is understudied in my opinion and to my knowledge this topic has not been subject to a scoping review such as the current one. Therefore, this review is timely and my lead to increased research in this area as it did a good job of pointing out future directions for research in the Discussion. The review was well-written and organized overall and did not have any fatal flaws. I think it will be of interest to readers of the journal and adds to the literature on the topic. Thus, I think the paper should eventually be published and I only have a set of very minor comments related to the writing, grammar, and organization of the paper. I believe the resolution of these issues the review should be published.

1.      Tables 1 and 3, there seems to be too much space between the lines. Please single space the table and see if there is anyway to left-justify the text. The spaces between some words make it difficult to read. Table 2 may have the second issue to a smaller extent.

2.      Line 312, should this sentence be its own paragraph or integrated into the prior paragraph. It is now a 1 sentence paragraph.

3.      I feel like the Conclusion section could be improved (lines 404 and after) being a little bit more detailed and longer.

4.      Line 58 should it be “aimed” please check other places in the text where past tense may have been better to use.

5.      Overall this paper was very well written with maybe only a few typos or minor grammatical errors. Line 81 I think there should be a comma after “but”.

6.      In the in text citations should the citations be put inside the period of the sentence? For instance, lines 34 and 36 and throughout the paper.

7.      The bibliography may have some mistakes. Some study titles are not capitalized and others have all the words capitalized. As just one example the first 6 references to each other. Some article titles the first letter of each word is capitalized some references all but the first word is not capitalized of the title.
